# Study on Interface Interaction between Uniaxial Geogrid Reinforcement and Soil Based on Tensile and Pull-Out Tests

**Xiaoguang Cai [1,2,3], Jiayu Feng [1], Sihan Li [1,2,3],\*, Honglu Xu [4], Weiwei Liu [5] and Xin Huang [1,2,3,4]**

1    College of Geological Engineering, Institute of Disaster Prevention, Sanhe 065201, China
2    Hebei Key Laboratory of Earthquake Disaster Prevention and Risk Assessment, Sanhe 065201, China
3    CEA Key Laboratory for Building Collapse Mechanism and Disaster Prevention, Sanhe 065201, China
4    Institute of Engineering Mechanics, China Earthquake Administration, Harbin 150080, China
5    Aolai Guoxin (Beijing) Testing Technology Co., Ltd., Beijing 101318, China
\*    Correspondence: lisihan@cidp.edu.cn; Tel.: +86-18730659598

**Abstract:** The interaction between reinforcement and soil is a key problem in the application of geosynthetics as reinforcement in geotechnical engineering. In this study, tensile and pull-out tests on a uniaxial geogrid were carried out using self-designed tensile and pull-out test equipment. The tensile test evaluated the tensile load–strain characteristics of a geogrid. Under the condition of lateral confinement, the tensile force and secant tensile stiffness of the geogrid increased with an increase in the normal stress when the strain was constant, and the secant tensile stiffness decreased with a decrease in the tensile rate. The stiffness coefficient was used to quantitatively describe the change in the stiffness of the reinforcement. Using the pull-out test, the variation laws of the pull-out force of the geogrid under different normal stresses and different longitudinal rib percentages were obtained. When the geogrid was broken, the pull-out force of the same type of geogrid was not significantly different under different normal stresses. With an increase in the longitudinal rib percentage, the pull-out force of the geogrid under the same normal stress gradually increased, and the apparent friction coefficient was obtained by analysis. The results of the apparent friction coefficient obtained by the analytical method in accordance with French specifications (NF P94-270-2020) are relatively safe compared to the experimental values.

**Keywords:** uniaxial geogrid; tensile test; pull-out test; tensile properties; friction coefficient

## 1. Introduction

In recent years, reinforced soil technology has been widely used in environmental protection, water conservancy, transportation, municipal, construction and other basic fields. In 2020, the proportion of geosynthetics in transportation infrastructure in China was 35.5%, with 31.1% in the field of water conservancy and hydropower, 13.8% in the field of environmental protection, and 19.6% in other fields. Yang Guangqing [1] determined that the composite growth rate of the market scale of China's geosynthetics industry was about 13.75% over the past five years, and predicts that the market scale of China's geosynthetics industry will be nearly CNY 80 billion by 2025. This indicates that reinforced soil technology must have broad application prospects. Reinforced soil technology shows good mechanical properties, structural seismic performance [2–9], and economic benefits [10]. The reinforcement effect of reinforced soil technology is achieved through the interaction between reinforcement and soil, which affects the safety and stability of the entire reinforced soil project [11]. Therefore, its interface characteristics are considered a key technical indicator.

The interaction mechanism of the reinforced soil interface varies with the different reinforced soil structures and the different interface positions of the same structure; the corresponding test methods are different as well [12]. E.M. Palmerira [13] proposed different test methods for different locations of slip surfaces (shown in Figure 1): Area A is the soil sliding on the surface of reinforcement using a direct shear test; Area B is the joint

deformation of the soil and reinforcement, the parameters of which are obtained by a tensile test in the soil; the shear between the soil and reinforcement in Area C is measured by a direct shear test of the inclined reinforcement; finally, Area D reinforcement is pulled out in the soil, and its parameters can be measured by the pull-out test.

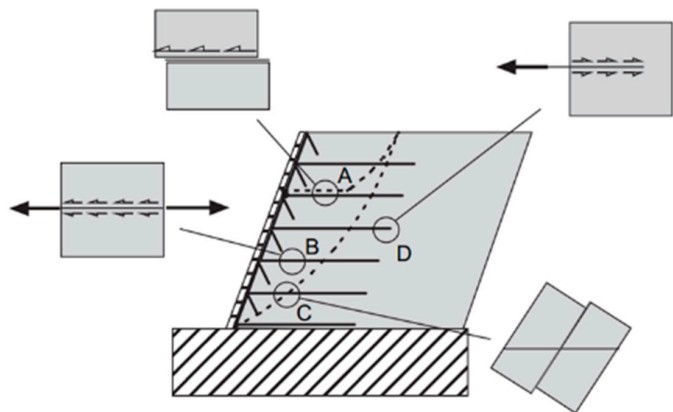

**Figure 1.** Internal mechanism of reinforced soil retaining wall [13].

Kokkalis and Papacharisis [14] used an improved direct shear test apparatus to conduct tensile tests of the geotextiles with lateral constraints. The elastic modulus and ultimate tensile strength increased with the increase in constraints. Juran and Christopher [15] carried out tensile tests of three materials (woven polyester strips, nonwoven geotextiles, and plastic geogrid) under confined and unconfined conditions, and found that the confined conditions had great influence on the material. The study of Wu and Tatsuoka [16] attributed the tensile properties of geosynthetics under constraint conditions to the coupling effect of constraint and friction on the interface between the reinforcement and the soil. McGown et al. [17] discussed the concept of "static interlocking". By incorporating soil particles smaller than the geogrid grid into the grid, this interlocking mechanism improves the interaction between the reinforcement and the soil, effectively improving the tensile performance. Mendes et al. [18] pointed out that under confined conditions, low strain, and high normal stress, the stiffness of geotextiles becomes higher, and the relative size of the gap between soil particles and geotextile affects its tensile properties. Therefore, in the design of reinforced soil structures, the influence of confined conditions on the mechanical parameters of reinforced materials should be fully considered. At present, the tensile test of reinforcement is often carried out in the air and then applied to reinforced soil engineering after the appropriate reduction, which is very different from the mechanical properties of reinforcement in soil.

The pull-out test is mainly used for the interaction between the reinforcement and the soil in the anchorage zone of the reinforced soil structure. By measuring the displacement and pull-out force of the reinforcement, the apparent friction coefficient between the reinforcement and the soil is obtained. A large number of experimental studies have shown that the mesh size of the reinforcement, the thickness of the filler, and the mesh size of the test box all affect the characteristics of the reinforced soil interface, and that the influence of various factors on the parameters of the reinforced soil interface has a certain regularity [19–22]. Of the 171 instable reinforced soil retaining walls investigated by Koerner et al. [4], 91% were reinforced with geogrids. According to national codes, in the calculation of internal stability, it is recommended that the value of the apparent friction coefficient be determined by the pull-out test. Although the recommended experimental value in the specification can be selected when the pull-out test is not carried out, there has been little research comparing this approach to the experimental values of the national codes. The numerical analysis method is another effective means of determining the stability of reinforced soil structures. However, in finite element analysis it is usually assumed that the grid is a continuous equivalent rough plane of reinforcement and that the interac-

tion of reinforcement depends on its shape and geometric characteristics. The estimation of the apparent friction coefficient of reinforcement in design becomes very complex.

At present, the research on the tensile properties of geosynthetics has mainly focused on geotextiles, and there has been little research on the tensile properties of geogrids under lateral confinement. There are different methods for obtaining the experimental value of the apparent friction coefficient in different countries' codes, but their rationality cannot be determined without comparison with experimental values. Therefore, in this study, we used the sand tensile test to compare and evaluate the tensile properties of reinforcement both under the condition of lateral confinement and without lateral confinement. The pull-out test was used to test the apparent friction coefficient under different normal stresses, and the experimental values in the national codes were compared in order to evaluate the rationality of the experimental value. In this study, experiments were mainly carried out on the possible reinforcement stress in the B and D zones.

## 2. Experimental Preparation

### 2.1. Testing Apparatus

Self-designed large tensile/pull-out test equipment [23] was used for the tensile and pull-out tests. Figure 2 shows a schematic diagram of the test equipment. A physical diagram of the test equipment is shown in Figure 3. The model box satisfied the test requirements of ASTM D6637 [24] and ASTM D6706 [25], with a size of 1.1 m (length) × 0.7 m (width) × 0.5 m (height). Three sides of the box were made of steel plate and one side of toughened glass. The vertical loading device was made of a rigid plate for loading. The normal stress, tensile force of the reinforcement, and displacement of the loading end were recorded by a pressure sensor, tension sensor, top rod displacement meter, and the Donghua data acquisition system. The fixture was composed of two steel plates fixed by bolts. In order to reduce the relative displacement between the geogrid and the fixture, two layers of rubber pads were placed on the contact surface of the geogrid and fixture. The control range of the pull-out rate was 0~250 mm/min, and the maximum tensile force was 5 t.

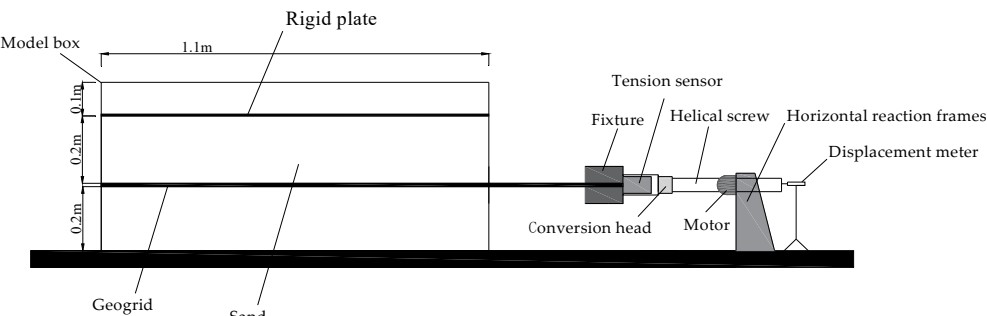

**Figure 2.** Diagram of test equipment.

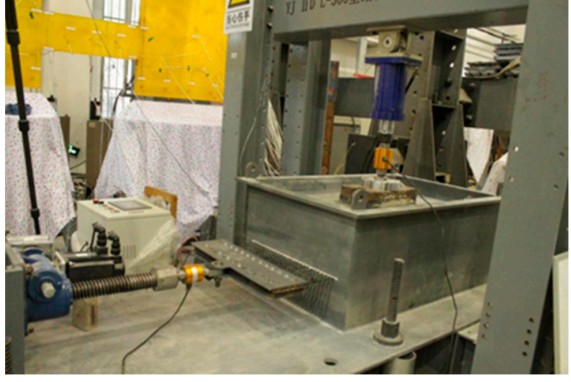

**Figure 3.** Physical view of test equipment.

### *2.2. Material Used*

#### 2.2.1. Sand

In this study, standard sand was used as backfill. The particle gradation curve and related physical and mechanical parameters are shown in Figure 4 and Table 1, respectively. The coefficient of curvature (Cc) and the uniformity coefficient (Cu) of the backfill were 1.262 and 2.055, respectively, indicating medium sand with poor gradation.

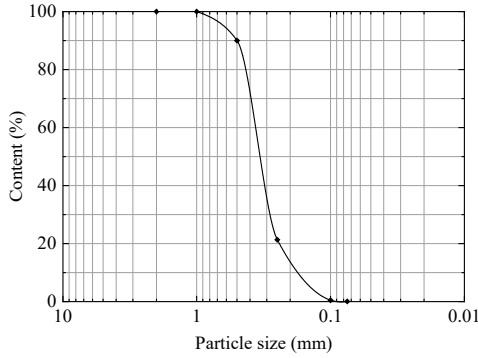

**Figure 4.** Particle gradation curve.

**Table 1.** Physical and mechanical parameters of sand.

| Relative Compaction (%) | Density $\rho$ (g·cm$^{-3}$) | Specific Gravity Gs | Angle of Internal Friction $\varphi$ (°) | Characteristic Particle Size (mm) | | |
|:---:|:---:|:---:|:---:|:---:|:---:|:---:|
| | | | | $d_{10}$ | $d_{30}$ | $d_{60}$ |
| 70 | 1.82 | 2.86 | 41 | 0.18 | 0.29 | 0.37 |

#### 2.2.2. Geogrid

In this study, three types of uniaxial geogrid were used for the test (G1: no rib removal treatment; G2: 60% of the longitudinal ribs remaining after rib removal; G3: 40% of the longitudinal ribs remaining after rib removal), as shown in Figure 5. The basic parameters of the uniaxial geogrids are shown in Table 2. The G1 geogrid was used for the tensile test to study the change in the stiffness and tensile ultimate strain under different normal stress and tensile rates. The G1, G2, and G3 geogrids were used for the pull-out test to study the interaction between the reinforcement and the soil interface.

**Table 2.** Basic parameters of uniaxial geogrids.

| Type | Thickness (mm) | Rib Spacing (mm) | Length of Tensile Unit (mm) | Raw Material |
|:---:|:---:|:---:|:---:|:---:|
| Uniaxial geogrid | 1 | 22.2 | 225 | Polyethylene |

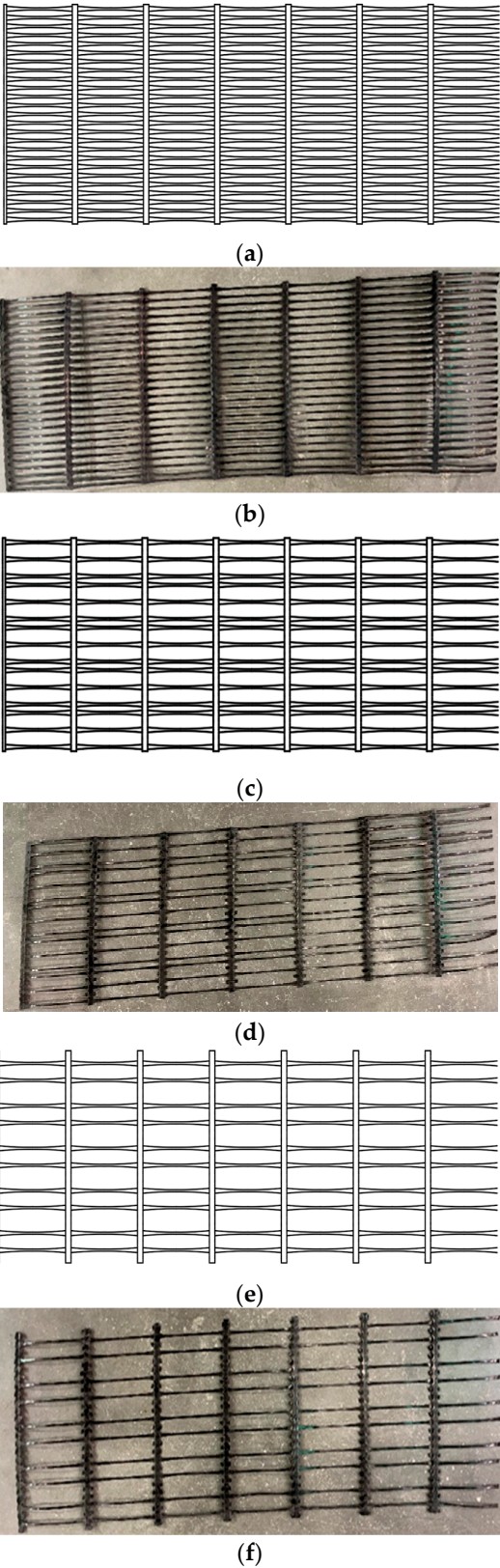

**Figure 5.** The three kinds of uniaxial geogrid specimens used in the tests: (**a**) G1 uniaxial geogrid (no rib removal treatment), (**b**) photo of G1 uniaxial geogrid, (**c**) G2 uniaxial geogrid (60% of longitudinal ribs remaining after rib removal), (**d**) photo of G2 uniaxial geogrid, (**e**) G3 uniaxial geogrid (40% of longitudinal ribs after rib removal), and (**f**) photo of G3 uniaxial geogrid.

### 2.3. Testing Program

Tensile tests in sand allow for full consideration of the effect of lateral restraint on the mechanical properties of the reinforced materials. The tensile test was carried out using the wide strip tensile method. According to the relevant provisions of the Test Specification for Geosynthetics in Highway Engineering (SL235-2012) [26], the tensile rate of the reinforcement was $(20 \pm 1)\%$/min of the nominal clamping length. A diagram of the tensile test is shown in Figure 6.

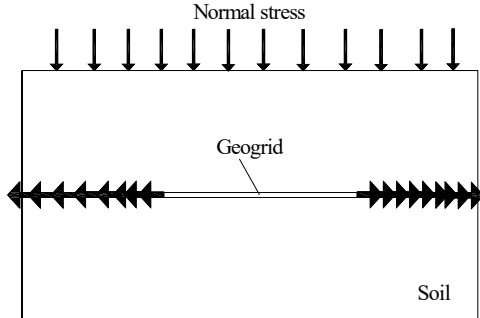

**Figure 6.** Diagram of the tensile test.

In the tensile test under unconfined conditions, the geogrid was first placed in the model box. The front end of the geogrid was tightly connected to the loading device by the fixture, and the rear end was fixed by the fixture, as shown in Figure 7. The sand samples with lateral constraints were prepared in the model box. The sand samples were divided into four layers. After each layer was filled, the sand samples were compacted and smoothed with a vibrator so that the relative density of each layer reached 70%. After the first two layers were filled, the geogrid was laid and then tightly connected to the loading section by a fixture. As shown in Figure 8, the tail was fixed by a fixture as well.

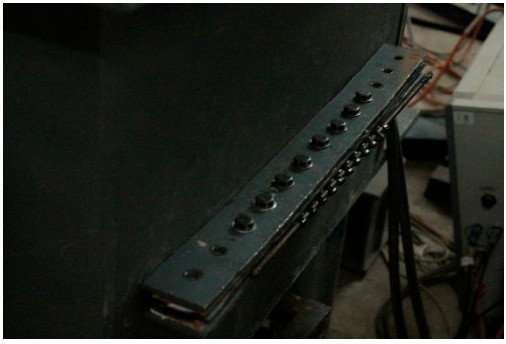

**Figure 7.** Tail fixture.

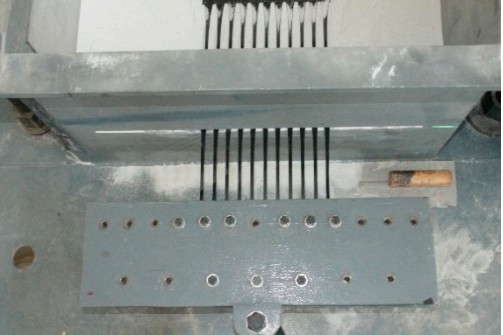

**Figure 8.** Head fixture.

The tensile test (TG1) of the G1 geogrid under unconfined conditions was carried out at a rate of 247 mm/min and the tensile–strain curve under unconfined conditions was obtained. The tensile test in sand, with $\sigma_v = 40$ kPa (TG2), $\sigma_v = 40$ kPa (TG3), and $\sigma_v = 60$ kPa (TG4), was used to study the influence of the tension–strain of the G1 reinforcement under confined conditions. Under the $\sigma_v = 40$ kPa condition, we changed the tensile rate to 200 mm/min (TG5), 150 mm/min (TG6), and 1 mm/min (TG7) in order to study the effect of the strain rate on the tension–strain of the G1 geogrid under confined conditions. The tensile test conditions are shown in Table 3.

**Table 3.** Tensile test condition table.

| Test Number | Grid Type | Length (mm) | Width (mm) | Normal Stress (kPa) | Rate (mm/min) |
|---|---|---|---|---|---|
| TG1 | G1 | 1400 | 240 | 0 | 247 |
| TG2 | G1 | 1400 | 240 | 20 | 247 |
| TG3 | G1 | 1400 | 240 | 40 | 247 |
| TG4 | G1 | 1400 | 240 | 60 | 247 |
| TG5 | G1 | 1400 | 240 | 40 | 200 |
| TG6 | G1 | 1400 | 240 | 40 | 150 |
| TG7 | G1 | 1400 | 240 | 40 | 1 |

The pull-out test was used to simulate the frictional resistance at the reinforcement–soil interface when the reinforcement material in the sand was pulled out under normal stress. The coefficient of friction between the reinforcement and the soil was determined by measuring the displacement of the reinforcement and the head tension of the reinforcement. A diagram of the pull-out test is shown in Figure 9. The pull-out rate was 1 mm/min, according to the ASTM specifications [25]. Then, pull-out tests of the G1, G2, and G3 geogrids under 20 kPa (PG1~PG3), 40 kPa (PG4~PG6), and 60 kPa (PG7~PG9) were carried out to study the influence of different normal stress and longitudinal rib percentage on the interface characteristics of reinforced soil. The tail was not fixed in the pull-out test, and both the production process of the sand samples and the layout of the geogrid were the same. The pull-out test conditions are shown in Table 4.

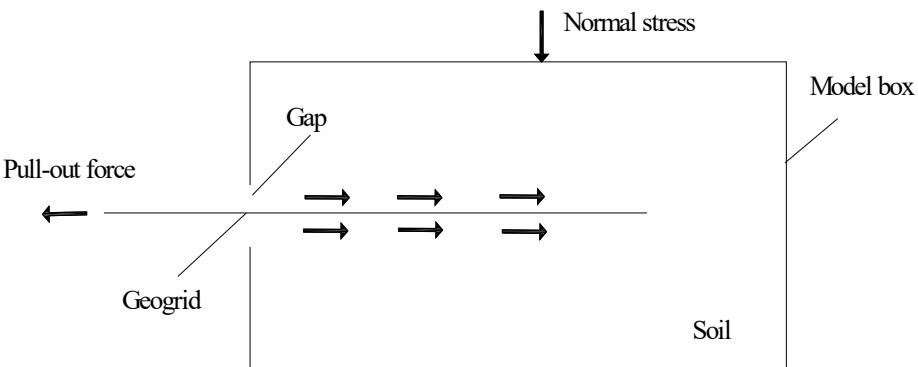

**Figure 9.** Diagram of the pull-out test.

**Table 4.** Pull-out test conditions.

| Test Number | Grid Type | Length (mm) | Width (mm) | Normal Stress (kPa) | Rate (mm/min) |
|---|---|---|---|---|---|
| PG1 | G1 | 1400 | 550 | 20 | 1 |
| PG2 | G2 | 1400 | 550 | 20 | 1 |
| PG3 | G3 | 1400 | 550 | 20 | 1 |
| PG4 | G1 | 1400 | 550 | 40 | 1 |
| PG5 | G2 | 1400 | 550 | 40 | 1 |
| PG6 | G3 | 1400 | 550 | 40 | 1 |
| PG7 | G1 | 1400 | 550 | 60 | 1 |
| PG8 | G2 | 1400 | 550 | 60 | 1 |
| PG9 | G3 | 1400 | 550 | 60 | 1 |

## 3. Results

Normal stress was applied to the upper part of the sand using the steel plate, although there is uncertainty regarding the normal stress applied to the geogrid. This may be due to the friction of the side wall of the model box, which leads to a decrease in the normal stress of the reinforcement.

In order to avoid the above problems, double-layer polyethylene is usually laid between the side wall of the model box and the sand, and the lubricating material is coated between the two layers of polyethylene [27]. Because the process was complicated and the test conditions were limited, the normal stress of the reinforcement was corrected through analysis and calculation of the Terzaghi formula (see Appendix A). The distribution of the normal stress in the sand is shown in Figure 10. Under normal stress conditions of 20 kPa, 40 kPa, and 60 kPa at the top, the corresponding normal stress of the reinforcement was 20.23 kPa, 37.33 kPa, and 54.43 kPa, respectively.

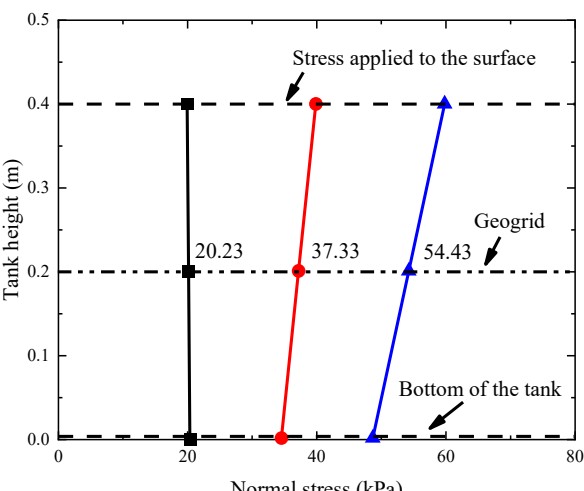

**Figure 10.** Relationship between normal stress and height of model box.

### 3.1. Tensile Test

#### 3.1.1. Effect of Normal Stress

Figure 6 shows the influence of normal stress on the tension–strain of the G1 geogrid. The strain is defined as the displacement of the loading end divided by the test length of the reinforcement (L = 1300 mm), expressed as a percentage. From Figure 11 it can be seen that, first, the strain of the geogrid is high under unconfined conditions. With the increase in normal stress, the strain of the geogrid decreases gradually, and the tensile force of the geogrid increases by 11–15% under confined conditions. (Second, under lateral confinement, the tensile force of the geogrid increases because the sand is embedded in the geogrid mesh hole, which limits the strain on the geogrid; additionally, the change in

the tensile force with strain is nonlinear under the conditions of lateral confinement. Third, with the increase in normal stress, increased tension is required for relative displacement between the transverse ribs of the geogrid and the sand, which is in keeping with the results of Balakrishnan et al. [28]. At the same time, the increase in the interlocking effect leads to an increase in the curvature of the tensile–strain curve of the geogrid, which decreases the strain rate of the geogrid with the increase in normal stress under the same tensile force. Fourth, according to the results of this tensile test, the tensile strength of the reinforcement in the sand is restricted by the strength of the reinforcement itself. As the strength of the geogrid is less than its tensile force in the sand, a fracture occurred at the junction of the transverse rib and the longitudinal rib, as shown in Figure 12.

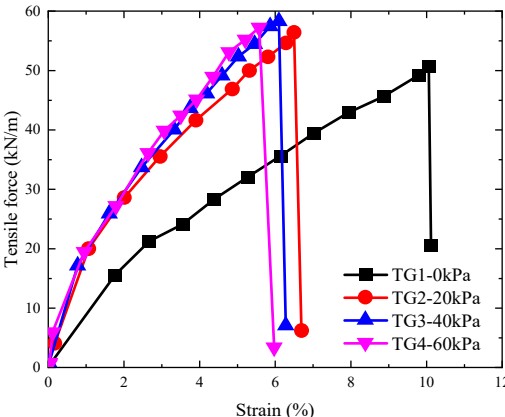

**Figure 11.** Effect of normal stress on tension–strain.

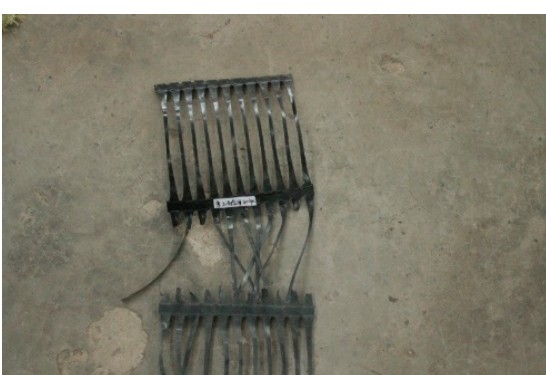

**Figure 12.** Uniaxial geogrid fracture diagram.

The confinement of the geogrid in sand affects the stiffness of the geogrid [18]. Therefore, the secant tensile stiffness–strain curve is shown in Figure 13. The secant tensile stiffness, $J_\varepsilon$ (kN/m), of reinforcement is calculated by the following formula:

$$J_\varepsilon = \frac{T}{\varepsilon} \tag{1}$$

where $T$ is the tensile force (kN/m) generated by the corresponding strain, $\varepsilon$ (%). In order to quantitatively describe the change in the secant tensile stiffness of the reinforcement in sand, the dimensionless factor stiffness coefficient $\delta$ is introduced, with the following formula used for calculation:

$$\delta_\varepsilon = \frac{(J_\varepsilon)_c}{(J_\varepsilon)_{un-c}} \tag{2}$$

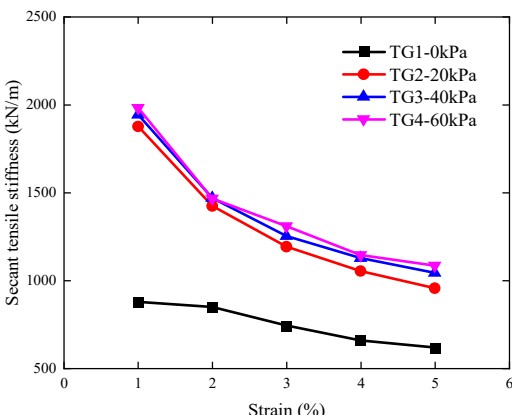

**Figure 13.** Effect of normal stress on stiffness–strain.

In the above formula, $(J_\varepsilon)_c$ is the value of the secant tensile stiffness tested under confined conditions and $(J_\varepsilon)_{un-c}$ is the value of the secant tensile stiffness tested under unconfined conditions. Table 5 summarizes the influence of normal stress on the stiffness coefficient under conditions of 2% strain, 5% strain, and peak strain,.

**Table 5.** Effect of normal stress on stiffness coefficient.

| Test Number | $\sigma_v$ (kPa) | $J_{0.02}$ | $J_{0.05}$ | $J_{peak}$ | $\delta_{0.02}$ | $\delta_{0.05}$ | $\delta_{peak}$ |
|---|---|---|---|---|---|---|---|
| TG1 | 0 | 847.62 | 616.75 | 503.15 | - | - | - |
| TG2 | 20 | 1423.03 | 955.20 | 867.44 | 1.68 | 1.55 | 1.72 |
| TG3 | 40 | 1468.42 | 1041.92 | 955.00 | 1.73 | 1.69 | 1.90 |
| TG4 | 60 | 1465.51 | 1082.78 | 1024.64 | 1.73 | 1.76 | 2.04 |

The change in the geogrid secant tensile stiffness is shown in Figure 13. The geogrid secant tensile stiffness under confined conditions was greater than that under unconfined conditions. The stiffness coefficient, $\delta$, represents the increase in tension under confined conditions. When the geogrid was subjected to lateral confinement and normal stresses of $\sigma_v$ = 20 kPa, 40 kPa, and 60 kPa were applied, the stiffness coefficients $\delta$ were 1.68, 1.73, and 1.73 at 2% strain and 1.55, 1.69, and 1.76 at 5% strain, respectively. It can be observed that the stiffness coefficient $\delta$ increased with the increase in normal stress. At 20 kPa and 40 kPa, the stiffness coefficient $\delta$ corresponding to the 2% strain of the geogrid increased, which may be due to the fact that, during the tensile process of reinforcement in sand, tension is provided by the relative displacement of the geogrid transverse ribs and the sand. With the increase in normal stress, the shear strength of sand increases while the secant tensile stiffness of reinforcement changes little. The stiffness of the reinforcement is mainly affected by the mutual displacement of the geogrid transverse ribs and the sand.

### 3.1.2. Effect of Loading Rate

Using the tensile test on the G1 geogrid under 40 kPa of normal stress at different tensile rates (247 mm/min, 200 mm/min, 150 mm/min, and 1 mm/min), we evaluated the effect of the tensile rate on the tension–strain of the reinforcement under lateral confinement. As shown in Figure 14, when the geogrid was stretched to 200 mm/min, the corresponding tensions with 2% strain and 5% strain were 28.10 kN/m and 48.05 kN/m, respectively. When the geogrid was stretched to 150 mm/min, the corresponding tension with 2% strain and 5% strain were 25.79 kN/m and 43.85 kN/m, respectively, and the corresponding tensile strength with 2% strain and 5% strain was 25.88 kN/m and 44.13 kN/m, respectively, at 1 mm/min. The tensile strength was slightly higher than 150 mm/min, and the secant tensile stiffness value was slightly higher. The higher the tensile rate, the less the sand particles can be rearranged, which leads to the force affecting the entire tensile test section. When the tensile rate is small, the sand is rearranged, and the force is not transmitted to the

back of the reinforcement test section. Formulas (1) and (2) were used to calculate TG3 and TG5~TG7. Overall, the decrease in the tensile rate leads to a decrease in the geogrid secant tensile stiffness. At 5% strain, the influence of the tensile rate on the geogrid tension and the secant tensile stiffness was small.

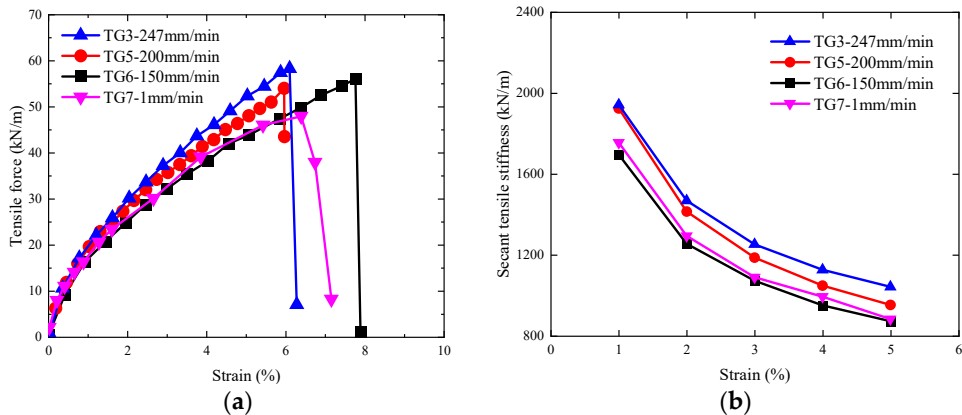

**Figure 14.** Effect of tensile rate on tensile properties of uniaxial geogrid; (**a**) tensile–strain curve, (**b**) stiffness–strain curve.

### 3.2. Pull-Out Test

#### 3.2.1. Effect of Normal Stress

Figure 15 shows the relationship between the pull-out force and the pull-out displacement of the G1, G2, and G3 geogrids during the pull-out test. The pull-out force of the geogrid is composed of the friction force of the longitudinal rib, the friction force of the transverse rib, and the passive resistance [29]. Under the action of different normal stress, the pull-out forces of the same type of geogrid were similar, and the pull-out curve decreased sharply after reaching the maximum pull-out force. The observation of the test phenomenon is that the geogrid broke, and the failure position was at the junction of the transverse rib and the longitudinal rib. The reason for this phenomenon is that the end bearing resistance of the transverse rib of the geogrid increased with the increase in the displacement of the pull-out end; as the strength of the junction of the transverse rib and the longitudinal rib was relatively low, tensile failure of the reinforcement occurred after a certain displacement was reached. Ziegler and Timmers [30] confirmed the same view. The uniaxial geogrid test results with different longitudinal rib percentages were compared. With an increase in the longitudinal rib percentage, the pull-out force increased, and the change in the pull-out force was significant.

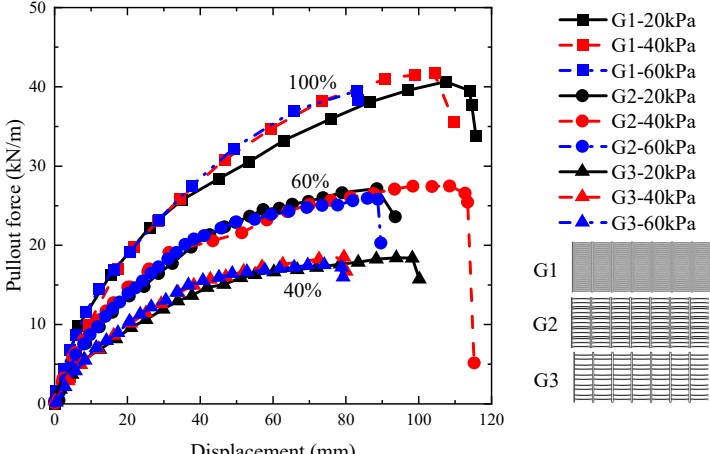

**Figure 15.** Relationship between pull-out force and displacement of three uniaxial geogrid specimens under different normal stresses.

In order to evaluate the proportion of pull-out force provided by the longitudinal and transverse ribs in the uniaxial geogrid, the pull-out force ($P_r$) of three types of geogrids was evaluated. The friction and passive resistance provided by the transverse ribs of the G1, G2, and G3 geogrids were considered to be the same, and the difference between them was that the friction provided by the longitudinal ribs was different. The end bearing resistance and friction ($P_{rT}$) of the transverse ribs were calculated by the following formula:

$$P_{rT} = P_{r(G2)} + P_{r(G3)} - P_{r(G1)} \tag{3}$$

The longitudinal rib friction ($P_{rL}$) is

$$P_{rL} = P_{r(G1)} - P_{rT} \tag{4}$$

The pull-out force provided by the transverse and longitudinal ribs under different normal stresses is summarized in Table 6.

**Table 6.** Proportion of transverse rib and longitudinal rib in terms of pull-out force.

| $\sigma_v$ (kPa) | $P_{r(G1)}$ (kN/m) | $P_{r(G2)}$ (kN/m) | $P_{r(G3)}$ (kN/m) | $P_{rT}$ (kN/m) | $P_{rT}/P_{r(G1)}$ (%) | $P_{rL}$ (kN/m) | $P_{rL}/P_{r(G1)}$ (%) |
|---|---|---|---|---|---|---|---|
| 20 | 37.70 | 27.11 | 18.36 | 7.77 | 20.61 | 29.93 | 79.39 |
| 40 | 41.77 | 27.45 | 18.46 | 4.14 | 9.91 | 37.63 | 90.09 |
| 60 | 39.46 | 25.89 | 17.48 | 3.91 | 9.91 | 35.55 | 90.09 |

It can be seen in Table 6 that the passive resistance of the uniaxial geogrid was small, and the pull-out force was mainly provided by friction.

### 3.2.2. Apparent Friction Coefficient

In order to analyze the internal stability of a reinforced soil structure, it is necessary to evaluate the tensile strength of the reinforcement in the anchorage zone. Its pull-out force was calculated by the following formula [31,32]:

$$P_R = 2L\sigma'_V f_b \tan \varphi' = 2L\sigma'_V f* = 2L\sigma'_V \alpha F* \tag{5}$$

The following was obtained:

$$f_b \tan \varphi = f* = \alpha F* \tag{6}$$

where $P_R$ is the pull-out force per unit width, L is the anchorage zone length of the reinforcement, $\sigma_v'$ is the effective normal stress, $\varphi$ is the internal friction angle of the sand, $f_b$ is the interaction coefficient of the reinforcement and the soil, $f*$ is the apparent friction coefficient, $F*$ is the pull-out force coefficient, and $\alpha$ is the scale effect correction factor considering the nonlinear stress reduction of extensible material in the embedded length.

When $f_b$ is not explicitly expressed to be used for calculation, and in order to avoid numerical analysis that is too complex, the apparent friction coefficient can be used for calculation, which depends on the normal stress. The formula is as follows:

$$f^* = \frac{P_R}{2L\sigma'_V} \tag{7}$$

In the FHWA [32], there is an experimental value for the $f^*$ of the geogrid, that is, $\alpha = 0.8$, $F^* = 2/3 \tan \varphi$; however, the use of this coefficient has a high requirement for the sand, which needs to meet the requirements of the FHWA [32], and the maximum value of $\varphi$ is 34°.

In the China Railway Subgrade Retaining Structure Design Code (TB10025-2019) [33], the experimental value of the apparent friction coefficient is 0.3–0.4. In the Highway Subgrade Design Code (JTG D30-2015) [34], the experimental value of the apparent friction

coefficient is provided according to the type of soil. The value of cohesive soil is 0.25–0.4, the value of sandy soil is 0.35–0.45, and the value of gravel soil is 0.4–0.5.

The French Code (NF P94-270-2020) [35] provides different values based on the absence of pull-out tests. The values of the apparent friction coefficient are calculated according to different depths. The formulas are as follows:

$$\begin{cases} f* = f_0^* \frac{h_0 - h_a}{h_0} + f_1^* \frac{h_a}{h_0} & h_a < h_0 \\ f* = f_1^* & h_a > h_0 \end{cases} \tag{8}$$

where $h_a$ is the average soil depth considered in the calculation, $h_0$ = 6 m, and $f_0^*$ is the apparent friction coefficient at the top. The calculation formula is as follows: $f_0^* = 1.1\left(\frac{\tan \varphi}{\tan 36°}\right)$, where $f_1^*$ is the apparent friction coefficient at depths of 6 m or below and the calculation formula is $f_1^* = 0.8 \tan \varphi$.

The apparent friction coefficients of the three types of geogrids were compared with the standard values, and the results are shown in Figure 16. First, it can be seen that the apparent friction coefficient of the three types of geogrids decreases with the increase in normal stress. When the normal stress was 20~60 kPa, the apparent friction coefficient $f*$ of the G1, G2, and G3 geogrids were reduced to varying degrees, decreasing from 0.91 to 0.33, from 0.61 to 0.21, and from 0.41 to 0.15, respectively. The reason for this phenomenon is that under the action of low normal stress, the sand of the geogrid experienced shear expansion during movement and the weight of the surrounding sand prevented the shear expansion effect, which caused an increase in the vertical stress on the geogrid. Compared to the values obtained from the test, the experimental values provided by the French specification are relatively lager. Especially with the increase in normal stress, the experimental values tended to be dangerous, although their variation laws were consistent. The values given by the FHWA specifications, that is, the railway subgrade retaining structure design and highway subgrade design, are conservative at 20 kPa. In the design of reinforced soil structure, the French specification overestimates the tensile force of the reinforcement, which can easily lead to excessive panel deformation due to insufficient tensile force of the structure, thereby reducing the durability of the reinforced soil structure.

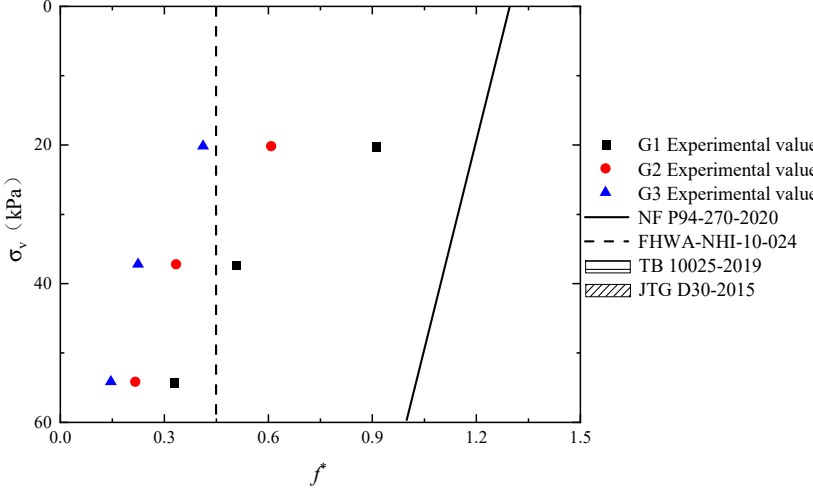

**Figure 16.** Comparison of test values and standard values of apparent friction coefficients.

## 4. Conclusions

In this study, tensile tests and pull-out tests were carried out on a unidirectional geogrid using tensile and pull-out test equipment. The effects of sand constraint, normal stress, and tensile rate on the tensile load–strain response of the reinforcement were analyzed. The stiffness coefficient was used to quantitatively describe the change in the tensile load–strain response of the geogrid. In the pull-out test, the influence of three geogrids

on the interface between the reinforcement and the soil under different normal stresses was analyzed, and the apparent friction coefficient was obtained. The experimental values provided in the specification were evaluated and compared to the national standards. The specific conclusions of this study are as follows.

(a) When the normal stress was 20 kPa, 40 kPa and 60 kPa, the secant tensile stiffness corresponding to a 5% strain increased by 1.55, 1.69, and 1.76 times, respectively, compared to that without lateral confinement. Due to the interlocking effect of the sand particles and the reinforcement, the secant tensile stiffness of the reinforcement under lateral confinement was higher than that without lateral confinement.

(b) With the decrease in the tensile rate, the secant tensile stiffness of the reinforcement decreased. Under confined conditions, the secant tensile stiffness of 200 mm/min, 150 mm/min, and 1 mm/min decreased by 0.92, 0.84, and 0.85 times, respectively, compared to the tensile rate required by the specification at a 5% strain, which may have been due to the rearrangement of the sand particles.

(c) With the increase in the percentage of the longitudinal ribs, the pull-out force of the geogrid increased gradually under the same normal stress, and the material itself broke down. Due to the occurrence of this failure mode, the increments of the pull-out force were small with the increase in normal stress.

(d) The apparent friction coefficient decreased with the increase in normal stress and increased with the increase in the longitudinal rib percentage under the same normal stress. While the change law was consistent with the change law of the French norms, the experimental value of the similar friction coefficient in the French Code is larger, which overestimates the tensile force of the reinforced soil structure and reduces its durability.

**Author Contributions:** Conceptualization, J.F.; writing—original draft preparation, X.C.; visualization, W.L.; data curation, S.L.; validation, X.H.; formal analysis, H.X. All authors have read and agreed to the published version of the manuscript.

**Funding:** This research was funded by the Fundamental Research Funds for the Central Universities, grant numbers ZY20215107 and ZY20220208, and the Langfang Science and Technology Support Plan Project, grant number 2021013172.

**Institutional Review Board Statement:** Not applicable.

**Informed Consent Statement:** Not applicable.

**Data Availability Statement:** Not applicable.

**Acknowledgments:** The writers appreciate Weiwei Liu, Honglu Xu, Chen Zhu, Baoshuang Jin, and Guanhao Shen for their help in the laboratory tests.

**Conflicts of Interest:** The authors declare no conflict of interest.

## Appendix A

The Terzaghi formula [36] can calculate the vertical stress ($\sigma_v$) along the soil column, assuming that the stress is proportional to the arch formed in the soil [37]:

$$\sigma_v = \frac{\gamma B_c}{2K_a \tan \phi_R}\left(1 - e^{-2K_a \tan \phi_R \frac{H_e}{B_c}}\right) + (\gamma(H_R - H_e) + \sigma_0)e^{-2K_a \tan \phi_R \frac{H_e}{B_c}} \qquad (A1)$$

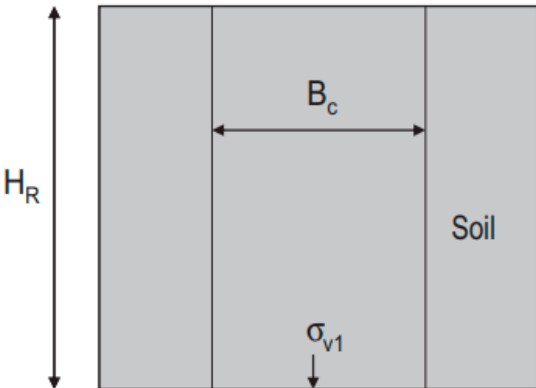

**Figure A1.** Terzaghi arching effect theory [38].

The parameter definitions and values are shown in Table A1.

**Table A1.** Parameters for the Terzaghi formula.

| Parameter | Definition | Figure |
|---|---|---|
| $K_a$ | Coefficient of active earth pressure, in this case $K_a = K_0$ | 0.34 |
| $B_C$ (m) | Width of soil column | 0.7 |
| $H_R$ (m) | Soil height | 0.4 |
| $H_e$ (m) | Equivalent soil settlement height | 0.4 |
| $\sigma_v$ (kPa) | Normal stress along soil column | - |
| $\sigma_{v1}$ (kPa) | Normal stress at the bottom of soil column | - |
| $\sigma_0$ (kPa) | Load | 0 |
| $\varphi_R$ (°) | Internal friction angle of soil | 41 |

For uniform settlement, $H_e = H_R$ [39], the normal stress applied to the bottom of the soil column can be calculated as follows:

$$\sigma_{v1} = \frac{\gamma B_C}{2K_a \tan \phi_R} \left( 1 - e^{-2K_a \tan \phi_R \frac{H_e}{B_C}} \right) + \sigma_0 e^{-2K_a \tan \phi_R \frac{H_e}{B_C}} \tag{A2}$$

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
