# Peer review of "Study on Interface Interaction between Uniaxial Geogrid Reinforcement and Soil Based on Tensile and Pull-Out Tests"

_sustainability, doi:10.3390/su141610386_

Round 1

Reviewer 1 Report

1 check the format of the references in the manuscript

2. move L53-54 to the final paragraph of the Introduction

3 Fig. 2, denote the name of test components instead of numbers

4. Fig 3, Chinese characters are found in the figure

5 change the description of soil to sand

6 Fig 4, give photos of the specimen

7 L132 describes the basic properties of geogrid

8 section 2.3, discuss the tensile and pull-out tests one by one and give their test diagram

9 L178-180, no clear what is the arch effect. Better discuss the test result only.

10. L187, not clear how to calculate.

11. L211 can not find what is “interlocking”, better discuss the observations only

12 L246 change tensile rate to loading rate

13, Table 4 is repeated by Fig. 9b

14, Eq 3 it is not clear how to calculate

15. Eq 5 confirm it is phai or phai’

16 Fig 11, Chinese characters are shown in the figure.

17 Appendix, name the figure and equations separately

Author Response

Dear Reviewer, thank you for taking time out of your busy schedule to review the manuscript. Your opinions are of great help to improve the quality of the manuscript. Now we have carefully corrected and replied the manuscript for this revision. The revision instructions are as follows.

Reviewer 2 Report

Major issues:

The authors should show some pictures of the tester and/or test setup.

Figures from 6 to 9 are incorrect when determine the values of the tensile strain (%) of the test curves, because the original length of the geogrid cannot be used. The geogrid length to determine the strain (%) is not the same as the original length when applying the confinement pressures (vertical pressures).

For that reason, the data from tensile tests cannot be used to evaluate geogrid properties including the secant tensile stiffness.

Conclusions (a) and (b) cannot be drawn from the test in this study and the test curves cannot be used because the tensile strain (%) determined in this study is incorrect. The manuscript should be prepared with better quality and more useful to readers.

However, Figure 10 for pull-out tests can be used to estimate interaction properties between soil and  reinforcement.

Other issues:

There are a lot of typo mistakes and Chinese words in this manuscript.

Figure 2 should be nicer.

It is reasonable to remove word “represents” in the note of Figure 2.

Figure 3 -  particle size distribution curve – should add inner vertical lines to show the log scale and remove Chinese words.

Table 1 should follow the journal guideline to show the units: (%), (mm), ..

Line 114: “drawing rate” should be “pull-out rate”

Line 133:  letter “s” - typo mistake.

Line 151: Check the unit (%) in “ (20% ± 1%) mm/min “

Line 152: “drawing test” should be “pull-out test”

Line 175: Table 2 Table of testing program” should be “Testing Program”

Explain why using the big change of the pull-out rate from 1 mm/min to 150, 200 and 247.

Line 193: For the tensile test, authors should show test pictures or some drawings of the tests.

Figure 11. Replace Chinese words

Please revise the text throughout the manuscript.

Author Response

(The authors gave the same response as above.)

Round 2

Reviewer 1 Report

The authors have revised the paper according to my comments. I have no comment now.

Author Response

Dear Reviewer, Thank you for taking time out of your busy schedule to review the manuscript. Your opinions are of great help to improve the quality of the manuscript. Now we have carefully corrected and replied the manuscript for this revision. The revision instructions are as follows.

Reviewer 2 Report

The revised manuscript has been improved. 

Author Response

(The authors gave the same response as above.)
